# Impact of a Digital Tool on Pharmacy Students’ Ability to Perform Medication Reviews: A Randomized Controlled Trial

**DOI:** 10.3390/healthcare11131968

**Published:** 2023-07-07

**Authors:** Armin Dabidian, Emina Obarcanin, Bushra Ali Sherazi, Sabina Schlottau, Holger Schwender, Stephanie Laeer

**Affiliations:** 1Institute of Clinical Pharmacy and Pharmacotherapy, Heinrich Heine University Duesseldorf, Universitaetsstrasse 1, 40225 Duesseldorf, Germany; e.obarcanin@hhu.de (E.O.); bushra.ali.sherazi@hhu.de (B.A.S.); sabina.schlottau@hhu.de (S.S.); stephanie.laeer@hhu.de (S.L.); 2Department of Pharmacy, National University Singapore, 18 Science Drive 4, Singapore 117559, Singapore; 3Institute of Pharmacy, Faculty of Pharmaceutical and Allied Health Sciences, Lahore College for Women University, Lahore 54000, Pakistan; 4Mathematical Institute, Heinrich Heine University Duesseldorf, Universitaetsstrasse 1, 40225 Duesseldorf, Germany; holger.schwender@hhu.de

**Keywords:** digital health, digital tool, medication review, pharmacy education, medication safety, pharmaceutical services, eHealth

## Abstract

Digital Medication Review Tools (DMRTs) are increasingly important in pharmacy practice. To ensure that young pharmacists are sufficiently competent to perform medication reviews after graduation, the introduction of DMRTs teaching in academic education is necessary. The aim of our study was to demonstrate the effect of DMRTs use on pharmacy students’ performance when conducting a medication review (MR) in a randomized controlled pre-post design. Forty-one pharmacy students were asked to complete a MR within 60 min, followed by a 10-min consultation with (intervention group) and without a DMRT (control group). The MR performance was subdivided into four categories: communication skills, subjective and objective patient data, assessment, and plan. Performance was assessed using objective structured clinical examinations (OSCEs) and analytical checklists. With the use of DMRTs, the overall performance was improved by 17.0% compared to the control group (*p* < 0.01). Improvement through DMRTs was seen in the subcategories “Assessment” and “Plan”. Furthermore, pharmacy students liked using DMRTs and felt more confident overall. Our study results demonstrate that DMRTs improve the performance of MRs, hence DMRTs should become an integral part of pharmacy curriculum. Consequently, digitally enabled pharmacists using DMRTs will be better prepared for their professional careers in pharmacy practice.

## 1. Introduction

In recent years, there has been a steady and dynamic change in the professional image of pharmacists in Germany, which has evolved from exclusively being a distributor of pharmaceuticals to becoming a key health care professional, with various consultation services in primary care [1,2]. Since 2022, the “On-Site Pharmacy Strengthening Act” allows pharmacists to offer selected “pharmaceutical care services to patients”, reimbursed by health insurances [1,3,4]. One core element of these services is the extended medication counseling for patients with polymedication [4] with ≥five systematic long-term medications/inhalants. The service includes pharmacists performing a brown-bag review [5] and a subsequent medication review, which can be provided and reimbursed once every 12 months [6]. If at least three acting medications/inhalants have been substituted, the service can also be reimbursed before the end of the 12-month period [6]. Pharmacists can independently initiate a medication analysis, without consulting a physician, if the above-mentioned legal criteria are met and if the patient has agreed to the service. The new pharmaceutical services profoundly increase pharmacists’ responsibility in primary health care, offering an in-depth medication service, outside the typical realm addressed by general practitioners. With this approach, the German health care system provides a foundation for enhanced medication safety, systematically integrating pharmacists as specialists to oversee complex medication regimens. Pharmacies in Germany have an estimated amount of one billion patient contacts per year and serve three million patients daily, therefore making use of pharmacists’ competencies and services is indispensable [7].

As Remelli et al. have demonstrated in their systematic review, almost two out of three type 2 diabetes mellitus (T2DM) patients are affected by polypharmacy [8]. Bauer et al. reported that T2DM patients in Germany are prescribed an average of 8.4 drugs [9]. Therefore, one important patient group that could benefit from medication counseling are T2DM patients. Pharmaceutical services in T2DM patients provided by pharmacists have already been shown to have benefits not only for patients, but also for health care systems by reducing the avoidable costs of T2DM [10,11,12].

A new trend of digital health and progressive digitalization has been steadily observed in German community pharmacies in recent years, with the aim of facilitating pharmacists’ daily work. For example, there are drug interaction alert software programs that inform pharmacists of potential drug interactions when drugs are being dispensed [13]. Digital competencies are being integrated into the practice of other pharmaceutical services, such as medication review tools, designed to assist pharmacists with performing medication reviews [14]. Studies have already demonstrated that performing medication review with similar tools identified more drug-related problems than without the use of such tools [15,16].

The rising popularity of digital health tools in everyday practice has led to numerous considerations to introduce the topic of digital health into the curriculum of health care professional students [17,18,19,20,21,22,23]. Since the use of digital health tools has been increasingly established among pharmacists for several years, there are already initial approaches or demands to implement digital health topics in the pharmacy curriculum in countries outside of Germany, especially in the United States and Canada [24,25,26,27,28,29]. To keep pace with recent digital healthcare developments and to prepare pharmacy students for the digital pharmaceutical care services, the Institute of Clinical Pharmacy and Pharmacotherapy Heinrich Heine University Duesseldorf introduced an elective practical course in mHealth and diabetes and digital health seminar. Additionally, its aim is to support a new generation of “digitally enabled pharmacists” [23].

The current study was designed to find out whether students can perform better by using a digital medication review tool compared to conducting medication reviews without such a tool. We decided to use the tool “MediCheck Education” (version 4.1.6). “MediCheck” is a well known commercial tool for German pharmacies, which is created to support medication review [30]. Furthermore, we also wanted to investigate the acceptance of digital tools among students in pharmacy education.

## 2. Materials and Methods

### 2.1. Participants and Study Design

The approval for this study was granted by the Ethics Committee of the Faculty of Medicine, Heinrich Heine University Duesseldorf. (Study number: “2022-1942-andere Forschung erstvotierend”). A total of 45 pharmacy students who were in their eighth and final semester at Heinrich Heine University Duesseldorf were considered for participation in the study. The study was conducted from May until June 2022 in the summer semester as a part of the clinical pharmacy course. Students were informed face-to-face by a clinical pharmacy lecturer during a seminar about the opportunity to participate in the study and were provided with a participation information sheet and a consent form, including a data protection statement. The inclusion criteria for the study were: signed informed consent form for participation in the study and signed data protection statement. Four students have already engaged with the medication review tool during a two-week internship prior to their eighth semester, thus they were excluded from the study. Voluntariness to participate in the study was emphasized and participation was not a requirement for passing the course. In order to preserve the confidentiality of personal data, participants were pseudonymized with study codes. Participants were randomly assigned to the intervention or control group. The overall study design is illustrated in Figure 1.

### 2.2. Procedure of First and Second OSCE

Performance of the participants in reviewing patient’s medication was assessed through first and second OSCEs. OSCE stands for “Objective Structured Clinical Examination” and is a modern verbal–practical examination method that has already been researched at the Institute of Clinical Pharmacy and is used as an innovative teaching and examination method [31,32,33,34]. OSCEs include about 20 stations, where one to two aspects of competence are tested per station. The present study, however, includes only one OSCE station, which deals with the conduct of a medication review. For reasons of linguistic comprehensibility, the term “OSCE” is used in the following instead of “single-station OSCE”.

In the first OSCE, the intervention and control group used the same analysis methods and did not use supportive digital medication review tools (DMRT). The participants received the respective case description, subjective and objective data of the patient, the patient’s medication schedule, and a form in which the participants could record their results. For research purposes, each participant was provided with a computer. Participants had access to a personal computer and were thus able to retrieve, for example, summaries of product characteristics of drugs and treatment guidelines. For both groups, the use of any DMRT was not allowed in the first OSCE. Each participant had exactly one hour to complete the medication review.

After the predetermined 60 min, participants were led to a room where they talked to an acting physician about the patient’s medication. In a role play scenario, physicians were played by two faculty members and an eighth semester student who did not participate in the study, whereas the study participants took on the role of the pharmacists. This conversation was documented and scored by a third person party, the OSCE examiner, using a standardized OSCE checklist. The OSCE examiners were all three clinical pharmacy faculty members and pharmacists qualified to rate the role play performance. The participants performed the OSCEs in a time-shifted manner (15-min time slots) to prevent possible exchange between the students, in the case that patient cases were the same. In addition, new patient cases were added for medication review after the first half of the day, throughout each day performing OSCE.

The student participants played the role of the pharmacists who informed the role-play physician about drug-related problems, drug interactions, and other errors regarding the patient’s medication. The physician–pharmacist consultation was limited to 10 min.

During the second OSCE, the procedure was identical; in terms of resources, the intervention and control groups differed. While the control group received the same equipment as in the first OSCE, the intervention group also used the DMRT “MediCheck Education” (by “pharma4u GmbH”, Eschborn, Germany) [30]. Both groups similarly had 60 min for the medication review and 10 min for the interview with the acting physician. To obtain the most objective assessment possible through the analytic checklist, the same examiner was always selected for each participant for the first and second OSCE. The study had a “blinded design”, and the student participants were not informed whether they belonged to the intervention group (working with “MediCheck Education”) or the control group (without “MediCheck Education”) until the beginning of the second OSCE. Throughout the study, neither the role-play physicians, nor the OSCE examiners, knew which participant belonged to intervention or control group. Prior to the OSCEs, both groups of role-play physicians and OSCE examiners were instructed on their respective tasks in order to be familiar with the procedure.

### 2.3. Patient Cases

For the first and second OSCE, four realistic patient cases were developed. It was considered that the cases were solvable within one hour. Three faculty members participated in the generation of the total of eight patient cases (SL, EO, and AD). To generate realistic patient cases, the anonymized patient cases from the EMDIA case series were used and modified accordingly [35]. The patient cases created were multimorbid elderly patients, all with T2DM. Other frequently occurring comorbidities were hypertension (*n* = 7) and dyslipidemia (*n* = 5). Overall, patients had a minimum of four and a maximum of six concomitant diseases, with some differences in diagnosis and medications. However, the patient cases were all kept at the same difficulty level. Moreover, it was ensured that for each patient case an equal number of participants from the control and intervention groups were allocated. In addition to eight OSCE patient cases, two further cases were designed for the “MediCheck Education” training. Each patient case is displayed in detail in Appendix A.

### 2.4. Software Tool “MediCheck Education”

“MediCheck” is a commercial web-based tool that can assist pharmacists in conducting medication reviews. Any web browser can be used to access the website, which requires a login. After logging in, an entry screen appears, which is divided into five categories and must be filled in with patient details for the medication review. The first category is called “Patient”, where the basic patient details, such as the patient’s age, height, and body weight are entered. The second category is called “Medication”. Here, all prescription and non-prescription medicines are recorded, including details of the package size, dosage, dosage regimen, dosage form, time of intake, and indication of the medicine. In the third category, problems and symptoms reported by the patient can be indicated. In the fourth category, all available laboratory and vital parameters of the patient are entered, such as cholesterol or blood sugar values from the last blood test. The fifth and last category includes the indication of diagnosed diseases or existing allergies of the patient. Once all five categories have been completed, the medication can be analysed. For the analysis, “MediCheck” uses an extensive database that contains, for example, different guidelines or product information on all medicines approved in Germany. This means, for instance, that an adjustment of the dose of metformin can be recommended in the case of existing renal insufficiency. The system checks for numerous parameters, such as drug–drug interactions, drug–disease interactions, contraindications, wrong dosages, and also the divisibility of tablets. After the analysis, all detected problems are listed and categorised in four risk categories with different colours. When listing all detected problems, the program lists sources and reasons as to why a problem was detected and additionally suggests corresponding alternatives. “MediCheck Education” and “MediCheck” do not differ in their functions; “MediCheck Education” is, however, merely used for training purposes.

### 2.5. “MediCheck Education” Training

Between the first and second OSCE, participants received training in DMRT “MediCheck Education”. Both control group and intervention group participants received this training simultaneously. By training both groups equally and at the same time, it was not revealed to participants whether they were in the intervention group or control group until the second OSCE appointment. The training lasted 2.5 h and was conducted online. Participants registered for the medication review tool without having to provide any personal information, as pseudonymized unlock codes were provided by the company. The pseudonymized activation codes ensured compliance with data protection requirements. After registration and login, participants were shown the user interface and general application options by an institute employee responsible for the entire study. The participants were then presented two fictitious patient cases for which they had to perform a medication review. For the processing of the patient cases, 45 min were calculated in each case. After the patient cases were reviewed, the results were discussed among all participants, as well as how “MediCheck Education” can be included in the medication review.

### 2.6. Measurement Instruments

#### 2.6.1. Analytical Checklist for OSCE

Participants’ performance was assessed through an analytic checklist. The checklists were individualized to each patient’s case, resulting in a total of eight different checklists. The same faculty members who created the patient cases (SL, EO, and AD) also created the respective checklists. The checklist always had the same structure, which was classified into four subcategories, namely, “Communication Skills”, “Patient data (subjective and objective)”, “Assessment”, and “Plan”. Due to the uniform structure of the checklist, it was easier for the OSCE examiners to adapt to a quick overview regarding the checklist. In addition to measuring the overall performance between groups, dividing the checklist into categories made it possible to conduct a subcategory analysis and thus to have a deeper insight for which areas the DMRT could be especially supportive. Each element was weighted equally. If an element was mentioned by the participant during the OSCE, the participant received one point. For each element, there was also a space for the OSCE examiners to take notes for any special occurrences or ambiguities. Since the patient cases were different, the checklists had different maximum scores. The checklist with the most items contained 29 points, whereas the checklist with the fewest contained 21. To create comparability between the different checklists, performance was measured as percentage. Since the checklists had different maximum scores, the subcategories were also scored differently, depending on the patient case. The subcategories “Communication Skills” and “Patient Data (subjective and objective)” had four points each for each patient case, whereas “Assessment” and “Plan” had different scores, depending on the patient case, and the measuring of performance was based on percentages, as well. The corresponding checklists for the patient cases are attached in Appendix A.

#### 2.6.2. Self-Assessment and Satisfaction Questionnaire

After the second OSCE, both the intervention and control groups were asked to complete a questionnaire with three questions about participants’ demographic data, such as age, sex, and previous professional experience in community pharmacies, as well as eight statements that the participants should rate. Regarding age, the option was given to select the age ranges “23 years or younger”, “24–30 years”, and “30 years or older”. For the sake of data protection, we decided against the disclosure of the exact age. For sex, there was a choice between “male”, “female”, and “diverse”. The options for answering the question of previous professional experience in pharmacies were “yes” and “no”. The eight statements were related to the participants’ self-assessment regarding the process of the OSCEs (6 statements) and the use of digital tools (2 statements) in teaching clinical pharmacy. The rating was carried out by a five-point Likert scale (1 = “Strongly agree”, 2 = “Agree” 3 = “Neither agree nor disagree”, 4 = “Disagree”, 5 = “Strongly disagree”). The statistical analysis of the statements was displayed by a forest plot. The eight statements are shown in Table 1.

### 2.7. Data Protection, Analysis, and Statistical Methods

All data collected by means of questionnaires and checklists were pseudonymized through the use of a code not indicating the names of the participants. During data analysis, a coding list existed that linked the names of the participants to the codes. The coding list was accessible only to the study director and project staff of the “Institute of Clinical Pharmacy and Pharmacotherapy at Heinrich Heine University”, Duesseldorf, and was destroyed after completion of the data analysis. After the destruction of the coding list, the data are available in anonymized form so that it is no longer possible to draw conclusions about individual participants. 

Randomization of participants into the intervention and control groups was performed using the “RAND function” by “Microsoft Excel” (version 2019). “Microsoft Excel” was used for data collection, and “RStudio” was used for data analysis.

A two-sided Mann-Whitney test was performed for the first OSCE for the comparison of performance between the intervention and control groups before using DMRT. For the second OSCE, a one-sided Mann-Whitney test was performed for the comparison of performance between the intervention and control groups after the use of the DMRT. To examine improvement in performance between the first and second OSCE within each group, a one-sided Wilcoxon signed-rank test was performed for the intervention group. Based on the hypothesis that the control group would not improve on the second OSCE, we performed a two-sided Wilcoxon signed-rank test on the control group. A significance level of α = 0.05 was considered in all statistical tests. 

The same statistical methods used for the overall data analysis were also employed for the respective subcategories. The aim was to be able to make a more precise statement about the field in which the performance of the participants changed if the performance changed between the first and second OSCE.

The questionnaire was analysed using the following method: After the demographic data was collected, it was analysed in percentage terms and presented in a table, divided into control and intervention group, representing demographic characteristics. The numbers from the Likert-scale for each of the eight statements of the questionnaire were analysed by calculating the arithmetic mean and the 95% confidence interval (CI). All means and 95% CIs were displayed in a forest plot. Consensus on a statement was reached when the 95% CI interval did not intersect the vertical line depicting “3” in the forest plot.

## 3. Results

### 3.1. Participants Characteristics

The response rate for recruitment was 100%, and all 41 eligible final-semester students signed the informed consent form and privacy policy agreement and were randomized into intervention and control group. Four students who did not participate in the study were scheduled to assist with the study. All 41 recruited participants attended all study appointments, including OSCEs before and after training with “MediCheck Education”. Randomization resulted in a group size of 21 participants for the intervention group and 20 participants for the control group. Table 2 describes demographic characteristics for control and intervention groups. Both the intervention and control group show very similar characteristics in terms of age, gender, and the number of those who had a previous professional experience in a community pharmacy.

### 3.2. Analytical Checklist—OSCE

#### 3.2.1. The Result of the Overall Performance

The performance assessment was conducted with an analytical checklist during both OSCEs. The performance scores are listed in Table 3. There was no significant difference (*p* = 0.784) between the intervention and control group at the first OSCE. After practicing with the DMRT “MediCheck Education” and using the tool in the second OSCE by the intervention group, the intervention group (52.1%) showed significantly better (*p* < 0.01) overall performance than the control group (35.1%). While the intervention group improved significantly (*p* < 0.05) in the second OSCE compared to the first OSCE, the control group deteriorated in the second OSCE when compared with the first OSCE (*p* < 0.05), as shown in Figure 2.

In our study, in addition to the overall performance of the groups, we evaluated the progress in the performance of medication reviews between the first and second OSCE. While significant progress regarding the performance could be observed in the intervention group, where the participants improved by a mean of 7.1% and a median of 8.9%, a negative performance development could be observed in the control group. The control group showed a deterioration of 8.4% in the mean and 5.6% in the median. Figure 3 shows the performance development of the participants in reviewing the patient’s medication between the first and the second OSCE.

#### 3.2.2. Subcategory Analysis

The performance of the participants’ medication review was recorded through an analytical checklist during the OSCEs. The checklist was divided into four categories. (1st “Communication Skills”, 2nd “Patient Data (subjective and objective)”, 3rd “Assessment” and 4th “Plan”). It was assessed in which areas the performance of the participants has improved through “MediCheck Education”. Table 4 shows the performance scores of the respective subcategories.

In the first OSCE, the control group and the intervention group did not differ significantly (e.g., *p* = 0.448 in the subcategory “Assessment” and *p* = 0.430 in the subcategory “Plan”) from each other in any of the subcategories. In the second OSCE, there was no significant difference in performance between the intervention and control groups in the categories of “Communication Skills” (*p* = 0.126) and “Patient Data (subjective and objective)” (*p* = 0.156). However, there were differences in performance in the categories “Assessment” and “Plan”, where the intervention group performed significantly (*p* < 0.01) better than the control group in the second OSCE.

In the “Assessment”, which deals with the identification of drug-related problems, such as dosing errors or drug interactions, the intervention group performed significantly better than the control group in the second OSCE. The category “Plan” focuses on solutions to problems that were either known and recorded in the category “Patient Data (subjective and objective)” or uncovered in the “Assessment”, such as incorrect dosing or occurring drug-related problems. The difference between the intervention and the control group in the first and second OSCE is shown graphically in Figure 4 for the subcategory “Assessment” and in Figure 5 for the subcategory “Plan”.

### 3.3. Self-Assessment and Satisfaction Questionnaire

Participants received questionnaires to provide statements about their self-assessment of performance during OSCEs and the use of digital tools in clinical pharmacy, as defined in Table 1. The analysis of the questionnaires was conducted using a forest plot, as shown in Figure 6. The statements were rated using a five-point Likert scale ranging from one (“strongly agree”) to five points (“strongly disagree”). Consensus on a statement was reached when the 95% confidence interval did not intersect the vertical line depicting “3” in the forest plot.

For half of the statements (statement number 1, 4, 7, and 8), there is consensus in both the intervention group and the control group. In particular, statements 7 and 8, which dealt with the use and meaningfulness of digital tools in general and their introduction into teaching, there was a strong agreement and support for implementation and establishment of digital tools to support medication reviews in teaching.

Regarding statements 2, 3, 5, and 6, the intervention group reached consensus, whereas the control group showed a similar tendency, but no consensus. Especially, the time aspect was rated very much as sufficient by the intervention group, and not by the control group. Statements 5 and 6, dealing with self-confidence during the conversation with the physician, showed that the intervention group felt more confident than the control group.

## 4. Discussion

In our randomized controlled trial, we found that pharmacy students using a DMRT performed better on medication review than students not using a digital tool. The performance of the DMRT-group was significantly better than the performance of the non-DMRT-group. The use of DMRTs helped pharmacy students with performing assessment, as well as creating the action plan for patients, whereas communication and collection of subjective and objective patient data performed equally with and without the digital support tool. In addition, the DMRT group’s satisfaction regarding the performance of the medication review, as well as the confidence to interact with the role play physician, were significantly greater than in the non-DMRT-group. 

To our knowledge, there are no studies investigating the performance of pharmacy students using DMRTs yet. However, some non-randomized studies with practising pharmacists are available and support our findings [27,28]. Curtain et al. demonstrated that pharmacists using medication review software for their medication review identified more drug-related problems than pharmacists without supporting software [15]. Verdoorn et al. investigated the impact of a clinical decision support system (CDSS) in a medication review [16]. In this study, a pre- to post-analysis of clinical medication review using a CDSS showed that the number of DRP identified increased. Both studies were retrospective. Our prospective RCT study design provides high quality evidence that DMRT use results in better outcomes of medication reviews. 

By further examining the subcategories of the medication review process, we were able to identify the categories where DMRT was most useful. Especially in the areas of assessing the medication prescribed and creating effective and safe medication plans, a digital assistant program proves to be a useful tool within the medication review process. This is in line with results from Curtain et al., emphasizing that software may help pharmacists in making good decisions and detecting important drug-related problems [36]. On the other hand, categories, such as communication skills and collection of all relevant subjective and objective patient data, were not affected by the digital tool. This is not surprising and can be expected, since DMRTs were not designed for these “soft skills”. 

In this study, DMRT was able to give pharmacy students more satisfaction and self-confidence. A major difference was the time required to complete the DMRT. The participants had one hour to solve each patient case; in contrast to the control group, the intervention group tended to agree that the time allotted was sufficient. In the studies by Levivien et al., Skalafouris et al., and Verdoorn et al., dealing with the use of digital tools in medication review, all emphasized the time-consuming factor that plays an important role in a medication review [16,37,38]. Verdoorn et al. even recommended to analyse the time aspect during a medication review in future studies. The different results between the DMRT-group and the non-DMRT-group, regarding the time aspect, could indicate that medication review is more efficient when using digital tools. However, it should be noted that this finding is based on a survey questionnaire, and the time for medication review was limited. Whether the intervention group carried out the medication review in a more time-efficient way was not analysed.

Our study may have some limitations. Firstly, the DMRT software reaches its limits when it comes to product characteristics of drugs or the implementation of current guidelines. As a result, digital tools cannot only create correct, but also incorrect correlations between drugs or suggest therapy recommendations that are no longer state of the art. Therefore, the use of digital tools in medication review is only warranted for pharmacists with clinical training and pharmacotherapy knowledge. Hence, our students received DMRT training (“MediCheck Education” training) and were trained to prioritize and solve drug-related problems. 

Secondly, the assessment of a participant’s performance was performed by an OSCE examiner. The OSCE examiner, however, could have some degree of subjectivity in the assessment and could have biased the results. These biases were addressed by strictly following the analytical checklists, as well as a uniform training for of all study staff evaluating the performance of the participants and grading the analytical checklist. Furthermore, the same examiner was always assigned to each participant in the first and second OSCE to prevent examiner-dependent variation in scoring. Moreover, care was taken to ensure that the individual items were clearly defined in the analytic checklist so that there was little room for interpretation by the examiners to assess performance.

Regarding the simulation of the physician role, the pharmacy faculty members and a student were also uniformly trained by the study author, and rehearsals of role plays were conducted. During those rehearsals, questions that arose were clarified, and clear guidelines were given in terms of communication. Again, before the first OSCE began, we discussed possible scenarios with participating role play physicians and analysed all patient cases in detail so that the interview would be as authentic as possible. Moreover, during the conversation with the study participants, the physicians in the role play were provided with documents, such as patient’s medication schedule and other fictitious data.

Our checklists were established in a two-step approach. First, two faculty members (AD and SS) tested the checklists on four pharmacy students as part of an internship in a small pilot study. These four students were not included in the later study. In this pilot study, the two faculty members investigated potential strengths and weaknesses of the checklists. Afterwards, the checklists were discussed with two other faculty members (EO and SL) who already had experience in creating OSCE checklists from previous studies [31,32,33], and the checklists were further optimised. Nevertheless, an assessment of the validity of our checklist by an external expert panel would have further strengthened our method.

When we analysed our results, we noticed that the non-DMRT-group performed worse in the review of patient’s medication in the second OSCE than in the first OSCE, which we had not anticipated before. While the DMRT-group, using the “MediCheck Education”, performed better and fulfilled more criteria on the analytical checklists, the analysis of our data showed that the non-DMRT-group deteriorated significantly. We suspect two causes that might be responsible for this observation: the first cause could be that participants in the non-DMRT-group were no longer as motivated in the second OSCE as they were in the first OSCE. Unlike participants of the DMRT-group, non-DMRT-group participants were not allowed to work with new methods to analyse patients’ medication. Another reason could be that the level of difficulty of the patient cases in the second OSCE was higher than in the first. The faculty members made sure that patient cases were approximately similar in terms of the number of diseases, as well as drugs, in order to generate patient cases of the same level of difficulty. Nevertheless, there is a possibility that the participants found the patient cases as a total more difficult in the second OSCE. Despite that, we ensured that approximately equal numbers of participants from the DMRT and non-DMRT groups were allocated to each patient case. However, in the event that the second case was deemed more difficult for the students to solve, it would make an even more positive statement in favour of using DMRT in medication review.

## 5. Conclusions

To the best of our knowledge, this is the first study to investigate the use of digital tools to support medication review in pharmacy students. Our study was able to demonstrate that pharmacy students do not only perform better in medication reviews when using “MediCheck Education”, but also feel more self-confident in the process. In addition, we could observe that students welcomed digital assistance and expressed their satisfaction with the introduction of digital tools in clinical pharmacy education. 

In our study, we observe that the use of digital tools requires training and clinical reasoning skills. DMRTs work best when they are assissted by trained medication experts—pharmacists—who are able to prioritize drug-related problems and provide correct recommendations in medication reviews.

Therefore, it is mandatory to introduce digital tools in teaching early, in a continuous effort to adapt pharmacy education to real-life pharmacy practice. The pharmacy curriculum at German universities does not require students to work in hospitals or pharmacies with real scenarios and real patients. Thus, an introduction of a course on the use of digital tools is even more important, so that, after graduation, students have at least acquired a theoretical knowledge in their daily work and have applied such tools on the basis of patient cases. However, further studies with larger sample size are needed to underline the establishment of digital tools in pharmacy teaching. 

## Figures and Tables

**Figure 1 healthcare-11-01968-f001:**
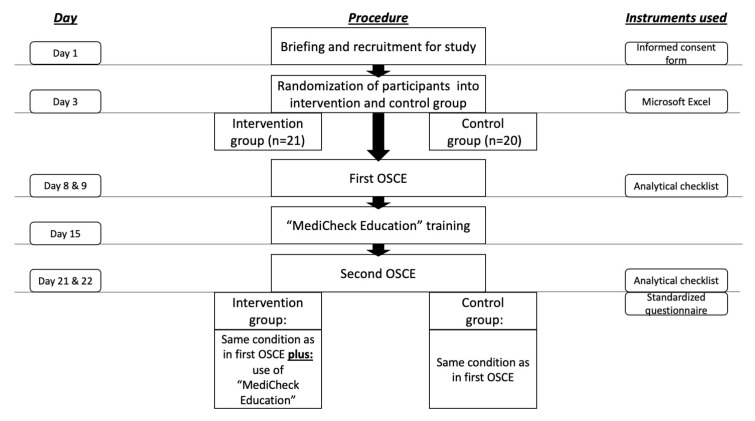
Study procedure and time-schedule. OSCE = objective structured clinical examination.

**Figure 2 healthcare-11-01968-f002:**
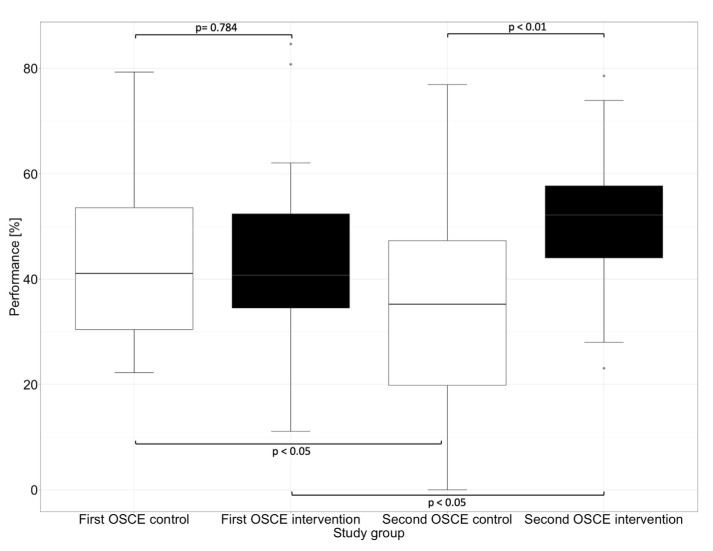
Overall performance in the first and second OSCE measured by an analytic checklist. Horizontal line = median; (*) = outlier.

**Figure 3 healthcare-11-01968-f003:**
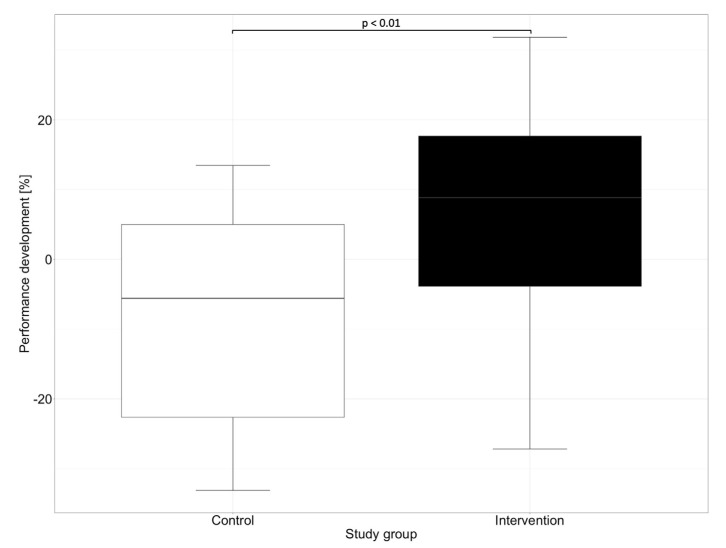
Performance development for the control and the intervention groups between the first and the second OSCE. Performance development was generated by subtracting participants’ performance on the second OSCE with their performance on the first OSCE. Horizontal line = median.

**Figure 4 healthcare-11-01968-f004:**
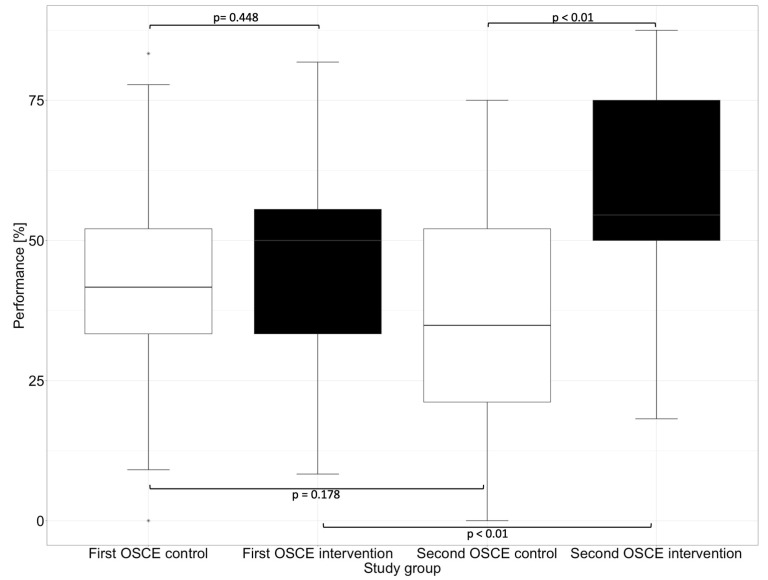
Performance of the first and second OSCE in the subcategory “Assessment”. Horizontal line = median; (*) = outlier.

**Figure 5 healthcare-11-01968-f005:**
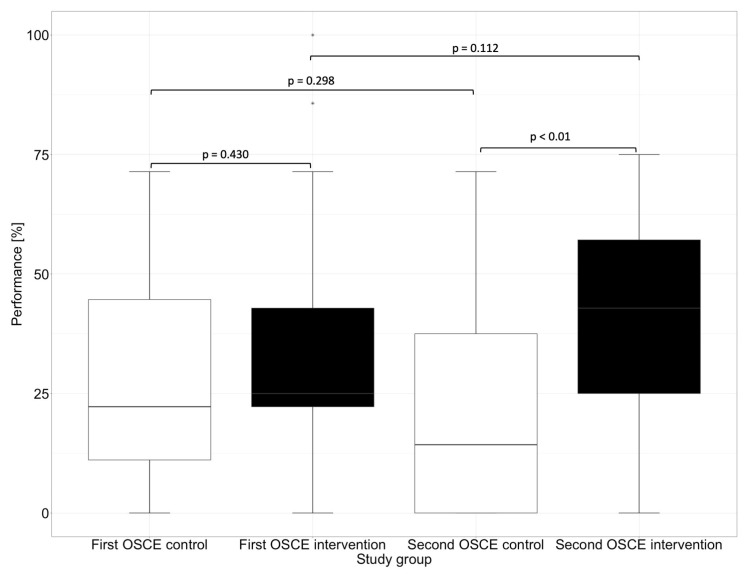
Performance of the first and second OSCE in the subcategory “Plan”. Horizontal line = median; (*) = outlier.

**Figure 6 healthcare-11-01968-f006:**
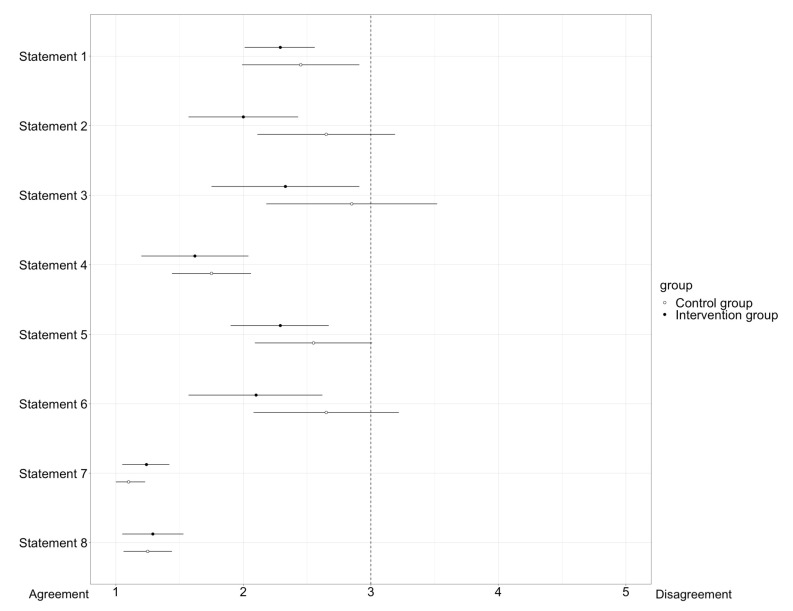
Display of the results of the questionnaire in a forest plot. The box represents the arithmetic mean. The horizontal lines to the left and right of the box indicate the 95% confidence interval. Consensus on a statement is reached when the confidence interval does not intersect the vertical line on 3 of the X-axis of the forest plot. While consensus was reached on all statements for the intervention group, no consensus was reached on statements 2, 3, 5, and 6 for the control group.

**Table 1 healthcare-11-01968-t001:** The eight statements of the participants’ questionnaire.

Statement 1	“I feel generally competent to perform a medication review”
Statement 2	“I feel more confident conducting the medication review today than I did during the medication review in the first OSCE”
Statement 3	“The time provided for the medication review was sufficient for me”
Statement 4	“The documentation provided for the medication review was sufficient for me (subjective and objective patient data, medication schedule, PC for research purposes, medication review tool if used)”
Statement 5	“I feel generally competent to have a face-to-face meeting with a physician”
Statement 6	“I feel more confident in today’s face-to-face meeting than I did in the face-to-face meeting in the last OSCE”
Statement 7	“The use of digital tools such as “MediCheck Education” is a useful addition for a medication review”
Statement 8	“The use of digital tools such as “MediCheck Education” is a useful addition to teaching in clinical pharmacy”

**Table 2 healthcare-11-01968-t002:** The demographic characteristics of the study participants.

	Intervention Group (*n* = 21) *n* (%)	Control Group (*n* = 20) *n* (%)
Age range		
23 years or younger	13 (62)	10 (50)
24–30 years old	5 (24)	7 (35)
30 years or older	3 (14)	3 (15)
Gender		
Female	16 (76)	14 (70)
Male	5 (24)	6 (30)
Previous professional experience (community pharmacy)		
Yes	5 (24)	5 (25)
No	16 (76)	15 (75)

**Table 3 healthcare-11-01968-t003:** Performance scores of the control and intervention groups in the first and second OSCE.

	Intervention Group (*n* = 21) Score in %	Control Group (*n* = 20) Score in %
First OSCE		
Mean (SD)	45.0 (18.1)	43.4 (15.8)
Median (MAD)	40.7 (16.5)	41.1 (17.7)
Second OSCE		
Mean (SD)	52.1 (13.4)	35.1 (19.7)
Median (MAD)	52.2 (12.1)	35.3 (21.9)

OSCE = objective structured clinical examination; SD = standard deviation, MAD = mean absolute deviation.

**Table 4 healthcare-11-01968-t004:** Performance scores of the first and second OSCE in four subcategories.

	Intervention Group (*n* = 21) Mean (SD) Score in %	Control Group (*n* = 20) Mean (SD) Score in %
Communication Skills		
First OSCE	52.4 (22.2)	57.5 (28.2)
Second OSCE	57.1 (25.2)	47.5 (29.1)
Patient data		
First OSCE	50.0 (22.3)	52.5 (21.3)
Second OSCE	47.6 (20.8)	38.8 (26.3)
Assessment		
First OSCE	45.2 (19.5)	41.9 (21.7)
Second OSCE	58.4 (18.0)	36.3 (22.8)
Plan		
First OSCE	33.2 (26.3)	27.1 (22.9)
Second OSCE	40.6 (20.0)	20.9 (22.5)

OSCE = Objective Structured Clinical Examination; SD = standard deviation.

## Data Availability

The dataset presented in this study is available from the corresponding author on reasonable request.

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
