# Peer review of "Impact of a Digital Tool on Pharmacy Students’ Ability to Perform Medication Reviews: A Randomized Controlled Trial"

_healthcare, 2023, doi:10.3390/healthcare11131968_

Round 1

Reviewer 1 Report

- The study design should be highlighted in the method.

- The authors stated that OSCE was used as an objective assessment method. Despite this, it is essential to recognize that a degree of subjectivity may still exist. A degree of subjectivity may be introduced into the evaluation procedure by variables such as the interpretation of performance, examiner bias, and variations in the difficulty of stations. 

- The authors mentioned that they have conducted before and after study, however, they did not do the necessary analysis for this study design, which are paired t-test (dependent sample t-test) /Wilcoxon test. They should do this analysis for the group that started without DMRT and then add it later on to see the impact of its addition. The current study design is not clear for me and is not presenting before and after study which ideally include one group. At the same time, it's not presenting two arms study as there is an addition for an intervention in the same group after a period of time (for the intervention group).

None.

Author Response

Dear Reviewer,

We are sending you our P2P-Reply as a Word document according to the template provided.

Thank you very much for reviewing our manuscript.

Reviewer 2 Report

Dabidian et al. performed a single-institution randomized controlled trial to study the effect of introducing digital medical review tools on the performance of pharmacy students in the OSCE examination. They report improved performance with the use of one of these tools (MediCheck Education”, and participants reported enhanced satisfaction with the use of such tools. The study will be of interest to those working in the pharmacy education research. The write up of the manuscript has several grammatical and typing errors that need to be carefully corrected before the next submission. In addition, I have several comments that I would like the authors will address before I can make the decision to endorse the manuscript for publication:

Introduction

Line 60: In my opinion, the whole paragraph about T2DM can cause confusion to the reader. The study at hand does not focus on T2DM. I would suggest the authors remove this paragraph and instead may add a short sentence indicating how T2DM patients are affected by polypharmacy, and how they may benefit from medication counseling offered by pharmacists.

Line 97: “introduced a one-of-a-kind elective practical course”. The “one-of-a-kind” expression seems over-exaggerated, unless the authors are welling to explain in detail why they described it using such expression.

Rationale: The expectation is that most pharmacy students receive training to use the DMRTs during the practicum part of the curriculum. The authors need to elaborate why they think that introducing a course on how to use these digital tools will be beneficial more than using these tools in real-case scenarios, for instance in hospital training. In addition, the authors indicate in the methods section, line 190, that training the students to use “MediCheck” lasted only two and a half hours, and in line 444 that that the training was “extensive”. If this is true, then introducing a semester-long course does not seem reasonably recommended.

Methodology:

Line 114: “Students were informed about the opportunity 114 to participate in the study and were provided with a participation information sheet and a consent form, including a data protection statement.”. How were the students recruited? Was this a face-to-face recruitment or an email recruitment or by other means? How were the students approached? How many were approached and how many agreed to participate? What is the response rate?

Minor comments:

Figure 1: Legend. Please correct the typo “procedure”.

Line 100: “A study was designed”. Please edit to “The current study was designed” or “This study was designed”.

Line 269: “The questionnaire was analysed the following”. Please correct the grammatical error in the sentence.

Line: 447: “such asassessment”. Please correct this typo.

I do not have access to the right similarity index tools to perform the plagiarism check.

Author Response

(The authors gave the same response as above.)

Reviewer 3 Report

This paper has a relevant and original purpose. Assessing the impact of training to perform Medication Reviews services is a useful answer for the health system.

The methodological proposal of students' randomization and  the simulated cases standardization seemed adequate to me. However, there is a methodological issue to be addressed to meet the title and objective proposals:

1) Title and objective proposal: measure the impact of training on the ability to carry out medication reviews. This measurement needs to be compared to a "gold standard"(or a Reference Standard)  analysis. And as a result of this comparison estimate a measure of agreement as a Kappa coefficient or another.

2) The data presented are only the participant's perception of their ability. This was not communicated in the title or objective. The perception of ability is not an useful information for the health care system.

So I strongly suggest Methods and Results modification.

Author Response

(The authors gave the same response as above.)

Reviewer 4 Report

Thanks for doing this project and sharing your results. Some comments which I hope will help you polish your final manuscript:

Methods

1) Please briefly explain what the DMRT tool does.  I am not familiar with it.  e.g., does it prompt the students as to what to include, such as are there any drug interactions, do any drugs need renal adjustments?  or does it actually provide information on the drugs such as identifying that metformin needs renal adjustments.

2) Are these performed based on student knowledge/memory or do students have access to any electronic references (e.g. Lexicomp or Micromedex?)

3) these comments are related to terminology. While not affecting the overall impact of your study, it is important to use terminology correctly.

- technically what you did was not an OSCE.  OSCEs generally consist of 10-15 stations, with each station testing a different skill.  What you did is an "OSCE station"

-OSCEs have very defined methodology.  OSCE station cases could be developed by a group of instructors, but the a panel of (outside/separate) experts would evaluate the checklist to make adjustments and assess overall face validity of the case and key.  A passing cutpoint for each case checklist would be developed by another group of experts based on criticality and importance.  There are other methods to set an OSCE cutpoint but you should specify whether such a  multistep process was used.  Using  OSCE methodology would add strength to your methods.  If the investigators themselves wrote the cases and keys without using an external panel, that weakens  your methods (but does not invalidate your results). A nonOSCE process should be mentioned in discussion under "limitations".

4) some of the methods related to randomization are duplicated (lines 253 but also 123)

5) another potential limitation to your study that should be mentioned is that only on grader assessed the intervention. If you had videotaped it and had a second person verify the score... that would have been helpful.  However, the detailed discussion of the assessors' training does strengthen your methods.

6) thanks for including a summary of your cases and checklists.  that helps the reader assess the level of difficulty of the cases and expectations for recommendations a pharmacist would make.  This is important for an international audience (i.e., would we all make such recommendations?) 

Results

I was wondering whether you needed both a table and a diagram for each.  It seemed like you could just include the table which included the p value and the standard deviation. 

In describing the results, you can summarize the table, without repeating everything that is reported in the table.  (e.g. both the intervention and control groups had similar numbers in each each group, gender and community pharmacy experience (see table 2)

Good luck!

minor English suggestions

line 424: "A major difference was the time required to complete the DMRT.  The Intervention group tended to agree that the time allotted was sufficient, while the control group did not. "

Appendix B, patient 1.  I think the word is supposed to be hypothyroidism?

Figure B3.  usually we say FEV1 has declined (rather than degraded)

Author Response

(The authors gave the same response as above.)

Reviewer 5 Report

With changing world perspectives this research paper is very significant in assessing pharmacy students competence regarding medication reviews using digital technology.

To make the medications safe for our patients this digital technology is definetely a breakthrough. We need to prepare students on how to use this sort of technology during their academic years. 

This method can be used to teach medical students in medical schools and we may conduct similar studies involving medical students.

Author Response

(The authors gave the same response as above.)

Round 2

Reviewer 2 Report

Thanks to the authors for responding to the comments and making the necessary changes to the manuscript.

I would suggest the authors include the discussion in their response to “Point 4” to the manuscript.

Best wishes

Author Response

Dear Reviewer,

Thank you for your feedback. We send you the P2P-Reply as a Word file in the corresponding template.

Kind regards

Reviewer 3 Report

Even though some suggestions of corrections were not made, the paper improved significantly with the performed corrections.

In Tables, I suggest including variable units only in the variables' description, removing unit repetitions from the body of tables.

Author Response

(The authors gave the same response as above.)
